# Biomimetic Theranostic Agents with Superior NIR-II Photoacoustic and Magnetic Resonance Imaging Performance for Targeted Photothermal Therapy of Prostate Cancer

**DOI:** 10.3390/pharmaceutics15061617

**Published:** 2023-05-30

**Authors:** Ling Liu, Shangpo Yang, Ziliang Zheng, Qingshuang Li, Chenchen Liu, Dehong Hu, Zhou Liu, Xiaoping Zhang, Ruiping Zhang, Duyang Gao

**Affiliations:** 1Department of Radiology, First Hospital of Shanxi Medical University, Shanxi Medical University, Taiyuan 030001, China; liuling1@sxmu.edu.cn (L.L.); zzlsxty@sxmu.edu.cn (Z.Z.); 2Paul C. Lauterbur Research Center for Biomedical Imaging, Institute of Biomedical and Health Engineering, Shenzhen Institute of Advanced Technology, Chinese Academy of Sciences, Shenzhen 518055, China; sp.yang@siat.ac.cn (S.Y.); qs.li1@siat.ac.cn (Q.L.); dh.hu@siat.ac.cn (D.H.); 3Department of Radiology, National Cancer Center/National Clinical Research Center for Cancer/Cancer Hospital & Shenzhen Hospital, Chinese Academy of Medical Sciences and Peking Union Medical College, Shenzhen 518116, China; zhou_liu8891@yeah.net; 4Department of Urology, Union Hospital, Tongji Medical College, Huazhong University of Science and Technology, Wuhan 430074, China; liucc@hust.edu.cn (C.L.); xzhang@hust.edu.cn (X.Z.)

**Keywords:** theranostic nanoparticles, photoacoustic imaging, the second near-infrared window, magnetic resonance imaging, photothermal therapy, cancer therapy

## Abstract

The accurate diagnosis and treatment of prostate cancer at an early stage is crucial to reduce mortality rates. However, the limited availability of theranostic agents with active tumor-targeting abilities hinders imaging sensitivity and therapeutic efficiency. To address this challenge, we have developed biomimetic cell membrane-modified Fe_2_O_3_ nanoclusters implanted in polypyrrole (CM-LFPP), achieving photoacoustic/magnetic resonance dual-modal imaging-guided photothermal therapy of prostate cancer. The CM-LFPP exhibits strong absorption in the second near-infrared window (NIR-II, 1000–1700 nm), showing high photothermal conversion efficiency of up to 78.7% under 1064 nm laser irradiation, excellent photoacoustic imaging capabilities, and good magnetic resonance imaging ability with a T2 relaxivity of up to 48.7 s^−1^ mM^−1^. Furthermore, the lipid encapsulation and biomimetic cell membrane modification enable CM-LFPP to actively target tumors, leading to a high signal-to-background ratio of ~30.2 for NIR-II photoacoustic imaging. Moreover, the biocompatible CM-LFPP enables low-dose (0.6 W cm^−2^) photothermal therapy of tumors under 1064 nm laser irradiation. This technology offers a promising theranostic agent with remarkable photothermal conversion efficiency in the NIR-II window, providing highly sensitive photoacoustic/magnetic resonance imaging-guided prostate cancer therapy.

## 1. Introduction

Prostate cancer (PC) is a prevalent malignancy and has emerged as one of the leading causes of cancer death in men worldwide in recent years [1,2]. The 5-year survival rate of patients diagnosed with PC at an early stage is significantly higher than that of patients diagnosed at a late stage, as late-diagnosed patients often have difficulty in receiving effective treatment [3,4,5]. Unfortunately, current medical imaging techniques, such as ultrasound and computed tomography, are difficult to accurately differentiate between normal tissue and early-stage lesion tissue. In addition, conventional blood tests for prostate-specific antigen (PSA) plays a crucial role in the screening of prostate cancer, while they cannot accurately identify the region of PC lesion tissue [5,6]. Therefore, there is a pressing need to develop highly sensitive imaging methods and techniques for the early detection of PC.

Magnetic resonance imaging (MRI) is a sophisticated non-invasive imaging modality that plays a critical role in the clinical diagnosis of PC due to its high spatial resolution and soft tissue imaging capabilities [7,8]. Contrast agents such as DTPA-Gd, Fe_2_O_3_ nanoparticles, MnO_2_ nanoparticles, etc., have been employed to augment the contrast and spatiotemporal resolution of MRI images [9,10,11,12,13]. Compared with Gd-based and Mn-based nanoparticles, Fe-based nanoparticles with low neurotoxicity and renal toxicity are adopted as effective contrast agents for enhancing T2-weighted MRI [13]. However, the low imaging sensitivity of MRI limits its use in the early diagnosis of PC [14]. Recent studies have shown that the fusion of MRI and optical imaging can effectively overcome the low sensitivity of MRI [15]. Photoacoustic (PA) imaging is a hybrid optical imaging technique that combines the benefits of high-sensitivity optical imaging and ultrasound imaging of deep tissue penetration and is an ideal complement to MRI [16,17]. In particular, the second near-infrared biological region (NIR-II, 1000–1700 nm) of PA imaging provides less background interference than the first near-infrared region (NIR-I, 700–950 nm) due to low tissue absorption and scatter [18,19,20,21]. Moreover, the contrast agents used for photoacoustic imaging can also be used as photothermal agents since they are involved in the process of converting light energy into heat energy [22,23,24,25,26,27,28,29]. Therefore, the fusion of MRI and NIR-II PA imaging could be an effective strategy to guide tumor therapy. The concept has been recently verified by constructing cancer cell membrane-coated theranostic agents [30,31,32,33,34]. Despite these progresses, the construction of multifunctional theranostic agents for the accurate diagnosis and therapy of PC is still a challenge.

Herein, we report a cell membrane-inspired nanoprobe for enhancing T2-weighted MRI and NIR-II PA signals for accurate diagnosis and imaging-guided efficient therapy of PC in a mouse model. The biomimetic nanoprobes were composed of Fe_2_O_3_ nanoclusters wrapped with polypyrrole and surface-modified with the PC cell membrane. This unique design simultaneously enhanced the T2 relaxation rate and absorption coefficient of the nanoprobes in the NIR-II biowindow and improved the molecular targeting of the nanoprobes [35,36]. Particularly, the nanoprobes can be used for efficient photothermal therapy (PTT) to kill PC cells under imaging guidance. The integrated strategy of MRI and NIR-II PA diagnosis and imaging-guided therapy provides opportunities for precise diagnosis and treatment of PC (Figure 1).

## 2. Materials and Methods

### 2.1. Materials

Iron chloride (98%), oleic acid (90%), 1-octadecene (90%), polyvinylpyrrolidone (Mw = 40,000, PVP40,000), and pyrrole were obtained from Sigma-Aldrich. The 1,2-dipalmitoyl-sn-glycero-3-phosphocholine (DPPC), 1,2-dioleoyl-sn-glycero-3-phosphocholine (DOPC), and cholesterol were received from Avanti. All commercial reagents were used as received without additional purification. Ultrapure water was used for the experiments.

### 2.2. Preparation of CM-LFPP

To prepare CM-LFPP (Cell Membrane-modified Lipids-Fe_2_O_3_-PVP nanoclusters implanted in Polypyrrole), oil-soluble Fe_2_O_3_ was first synthesized following the procedure in a previous report [37]. The oil-soluble Fe_2_O_3_ nanoparticles were then transferred into water using PVP40,000. Specifically, 40 mg of oil-soluble Fe_2_O_3_ nanoparticles and 0.8 g of PVP40,000 were dissolved in CHCl_3_, and the solvent was removed using a rotary evaporator. The obtained thin film was dispersed in water through ultrasonication, which was further conducted using an ultrasonic probe under an ice bath. The mixture was filtered by a 220 nm membrane to obtain PVP-modified Fe_2_O_3_ nanoclusters (FP).

The PVP-modified Fe_2_O_3_ nanoclusters were diluted with water to make the PVP mass ratio 4%. Subsequently, pyrrole (51.7 μL, 0.74 mmol) was added to the mixture. To initiate the reaction, FeCl_3_ (40 mg mL^−1^) was slowly dropped into the mixture at 37 °C, and the reaction was stopped after 24 h. The initiator and other reactants were removed by ultrafiltration, obtaining purified polypyrrole (PPY)-PVP-Fe_2_O_3_ nanoclusters (FPP).

Cell membrane fragments were extracted from PC-3 prostate cancer cells according to our previous method [38]. Lipids encapsulated FPP (LFPP) were obtained using the thin-film hydration method. DPPC, cholesterol, and DOPC (75:5:20) were dissolved in CHCl_3_, forming a thin film after rotary evaporation. The FPP solution was added to the dried phospholipid film, and the mixture was subjected to five cycles of freezing and thawing. The mixture was then extruded through a filter membrane with a pore size of 200 nm. Once LFPP was formed, cell membrane fragments were added to the solution at a ratio of 1:400. The mixture underwent five cycles of freezing and thawing, alternating between liquid nitrogen and a water bath at 65 °C, before being extruded (200 nm) to form CM-LFPP.

### 2.3. Characterization of the CM-LFPP

The absorption spectra of FP, FPP, and CM-LFPP were recorded using a PerkinElmer Lambda 750 UV-vis–NIR spectrophotometer. The zeta potentials and hydrodynamic diameters of LFPP, CM-LFPP, and cell membrane (CM) were measured using the Zetasizer Nano ZS. The morphology and elemental distribution of CM-LFPP were characterized by dropping it onto a carbon-coated grid and then analyzing it using transmission electron microscopy (TEM) after drying. The weight of nanoparticles in the dispersions was measured according to this procedure. An amount of 1 mL of the CM-LFPP solution was lyophilized using vacuum freezing and drying technology, and the resulting powder was weighed to determine the concentration of the nanoparticles. To compare the T2 relaxivity of CM-LFPP with the previous reports, we measured the amount of iron (Fe) element using ICP-MS after digesting the CM-LFPP.

### 2.4. Photothermal Performance of the CM-LFPP

To investigate the dose-dependent behavior, solutions of CM-LFPP with concentrations of 15.6, 31.3, 62.5, and 125 μg mL^−1^, as well as a PBS control, were irradiated with a 1064 nm laser at a power density of 0.6 W cm^−2^ for 360 s. To investigate the power-dependent behavior, a CM-LFPP solution with a concentration of 125 μg mL^−1^ was exposed to a 1064 nm laser with varying power densities (0.1, 0.2, 0.4, 0.6, and 0.8 W cm^−2^). Temperature variations were recorded at different time points using a thermal imaging camera (Ti400, Fluke, Everett, WA, USA). Temperature variation curves were also recorded with the laser both on and off, in order to evaluate the photothermal conversion efficiency of the CM-LFPP using the following equation [39]:(1)$$\eta =\frac{m_{D}c_{D}\left(T_{max}-T_{amb}\right)-Q_{dis}}{\tau_{s}I(1-10^{-A_{1064}})}$$

The mass of the CM-LFPP solution is denoted as *m_D_*, while *c_D_* represents the heat capacity of solvents. *τ_s_* represents the time constant of the CM-LFPP system, and *T_max_* and *T_amb_* denote the maximum equilibrium temperature and ambient temperature of the CM-LFPP, respectively, when exposed to the 1064 nm laser. *Q_dis_* refers to the amount of heat energy dissipated from the light absorbed by the solvents, while I represents the power density. Additionally, A represents the absorbance of the CM-LFPP at 1064 nm. The value of *τ_s_* is determined by using the following equation: (2)$$\tau_{s}=-ln\frac{T_{amb}-T}{T_{amb}-T_{max}}$$

In the context of in vivo PTT for cancer treatment, the tumor was intratumorally injected with both PBS and CM-LFPP, followed by exposure to the laser for a duration of 6 min. During this process, the temperature variation at different time points was recorded using a thermal imaging camera (Ti400, Fluke, USA).

### 2.5. Photoacoustic Imaging Ability of the CM-LFPP

The CM-LFPP’s photoacoustic imaging ability was assessed using a commercial 3D photoacoustic imaging system (Tomowave LOIS-3D). Photoacoustic signals of CM-LFPP solutions with varying concentrations (0, 0.063, 0.125, 0.25, 0.5, and 1 mg mL^−1^) were detected upon excitation by a 1064 nm laser. For in vivo photoacoustic imaging, images were recorded at different time points following intratumoral (0.1 mg kg^−1^) or intravenous (0.5 mg kg^−1^) injection of the LFPP/CM-LFPP, and the quantitative signal-to-background ratio (SBR) was analyzed from the images. The quantitative photoacoustic imaging data were obtained through a plugin installed in the 3D slice, developed by the company utilizing ImageJ.

### 2.6. Cell Culture and Cell Cytotoxicity

To evaluate the cytotoxicity of CM-LFPP, both bend.3 cell lines and prostate cancer PC-3 cell lines were used in an MTT assay. Normal and cancer cells were incubated at 37 °C with 5% CO_2_ in DMEM and seeded onto 96-well plates at a density of 5 × 10^3^ cells/well (100 µL). After removal of the original medium, DMEM containing various concentrations of CM-LFPP (0, 15.6, 31.3, 62.5, 125, 250, and 500 µg mL^−1^) were incubated with the cells. The cytotoxicity of CM-LFPP was measured using the MTT assay after 12 and 24 h of incubation. In addition, the PC-3 cell cytotoxicity of four groups including PBS-treated, CM-LFPP-treated, laser-treated, and CM-LFPP + laser- treated were also measured using the MTT assay. For CM-LFPP + laser- treated group, PC-3 cells were incubated with CM-LFPP (250 µg mL^−1^) for 4 h at 37 °C, washed with PBS, and irradiated under 1064 nm laser (0.6 W cm^−2^, 6 min).

### 2.7. MRI Capacity of the CM-LFPP

To assess the MRI imaging capacity of CM-LFPP, solutions with varying concentrations (c_Fe_ = 0, 0.016, 0.031, 0.063, 0.125, 0.25, 0.5 mM) were immersed in a water tank, and T2 relaxation time and T2-weighted images were obtained using a 3.0-T MRI scanning system (uMR790, Shanghai United Imaging Healthcare, China) with a head coil. For in vivo MRI, control images were obtained before the injection of contrast agents (LFPP or CM-LFPP) using the same system with a small animal coil. The MRI capacity was evaluated by obtaining MR images at 0.5 h post intratumoral injection of LFPP and CM-LFPP, respectively. Additionally, active tumor-targeting ability was investigated by acquiring MR images 24 h post intravenous injection of LFPP and CM-LFPP. The MRI was quantitatively analyzed by measuring the tumor signals of various slices using the MRI system’s software.

### 2.8. Animal Studies

The animal experiments were approved by the Animal Care and Use Committee of the Shenzhen Institutes of Advanced Technology, Chinese Academy of Sciences. Male BALB/c nude mice (20–22 g) were obtained from Beijing Vital River Laboratory Animal Technology Co., Ltd. (Beijing, China) and were adopted to construct the PC-3 tumor-bearing subcutaneous tumor model by subcutaneously injecting 5 × 10^6^ PC-3 cancer cells per mouse. The in vivo experiments were conducted when the tumor had grown to a size of approximately 100 mm^3^.

### 2.9. Hemolysis Test

Blood was collected from BALB/c mice weighing 20–22 g using anticoagulation tubes. The blood was washed with PBS (5 mL) and then centrifuged (2000 rpm, 3 min). Different concentrations of CM-LFPP were added to the samples, and a positive control with double-distilled water and a negative control with PBS were included. All samples were incubated at 37 °C for 1 h, and images were taken and the absorption at 540 nm was measured for each group. The hemolysis rate was calculated using the following formula. Hemolysis rate = (A_540-experimental group_ − A_540-sample_ − A_540-negative control_)/(A_540-positive control_ − A_540-negative control_) × 100%

### 2.10. Histological and Blood Biochemistry Analysis

To assess the safety of CM-LFPP, we collected the main organs of BALB/c mice treated with PBS and CM-LFPP and performed an H&E analysis. Additionally, we collected blood samples from mice with various treatments (PBS-treated, 1-day, 4-day, and 7-day after intravenous injection of CM-LFPP) and conducted biochemical analysis.

## 3. Results and Discussion

### 3.1. Synthesis and Characterization of the CM-LFPP

The biomimetic dual-modal nanoprobes (CM-LFPP) were designed and prepared as illustrated in Figure 1A. Initially, oil-soluble Fe_2_O_3_ nanoparticles were synthesized according to a previous report, exhibiting a uniform size with a diameter of approximately 5 nm [37]. Subsequently, the Fe_2_O_3_ nanoparticles were transferred into an aqueous solution by coating their surface with PVP, forming PVP-Fe_2_O_3_ nanoclusters. These nanoclusters were then embedded into polypyrrole (PPY) through a polymerization reaction. Finally, PPY-PVP-Fe_2_O_3_ was modified with lipids before insertion of the prostate cancer cell membrane. Transmission electron microscopy (TEM) images revealed that the nanoparticles formed clusters and were encapsulated inside the biomimetic lipids to yield CM-LFPP. The CM-LFPP exhibited a core-shell structure with a diameter of approximately 134 nm (dark core, FPP clusters, approximately 128 nm; light shell, cell membrane fragments, and lipid, approximately 6 nm) (Figure 1B). High-angle annular dark-field scanning transmission electron microscopy (HAADF-STEM) and elemental mapping were further conducted to investigate the structure of the CM-LFPP. The images showed that the Fe element was distributed in the core region, while the C, N, and O elements were distributed in both the core and shell regions (Figure 1C–H). This confirmed the core-shell structure of the CM-LFPP and verified the Fe_2_O_3_ clusters as the core of the CM-LFPP. The hydrodynamic diameter of the FPP was measured to be 138.3 ± 0.6 nm, which increased to 154.8 ± 0.9 nm after lipid encapsulation (LFPP). It further changed to 146.8 ± 1.3 nm (PDI = 0.241 ± 0.043) after being cloaked with the PC cell membrane (CM-LFPP), which was slightly smaller than that of the LFPP (Figure 1I). This may be due to the incorporation of cell membrane fragments into the lipid shell after successive extrusion, which was consistent with previous reports [38]. The zeta potential of the LFPP decreased from −12.9 ± 0.23 mV to −20.2 ± 1.01 mV after biomimetic modification, demonstrating that the cell membrane was incorporated into the LFPP (Figure 1J). Furthermore, we investigated the UV-Vis-NIR spectra of FP, Pyrrole, FPP, and CM-LFPP. The absorptions of the FPP and CM-LFPP ranging from 530 to 1100 nm were significantly increased compared to those of FP and Pyrrole, making them promising PA imaging agents and photothermal agents. This also demonstrated that the broad absorption of the FPP and CM-LFPP was likely due to the formation of PPY. Overall, the results suggest that the CM-LFPP nanoprobes possess excellent potential for use in dual-modal imaging and PTT for prostate cancer.

### 3.2. In Vitro Photothermal Performance of CM-LFPP

The strong absorption of CM-LFPP in the NIR-II region makes it a promising photothermal agent for converting photo energy into heat energy. Therefore, we investigated the in vitro photothermal conversion ability of CM-LFPP under 1064 nm laser irradiation. We captured infrared thermal images of the CM-LFPP solution (125 μg mL^−1^) at different time points and power densities. It indicates that the photothermal behavior of CM-LFPP is dependent on both time and power density. Specifically, we observed a more significant temperature variation at longer irradiation times and higher power densities (Figure 2A). Notably, at power densities of 0.6 W cm^−2^ and 0.8 W cm^−2^, the temperature of the CM-LFPP solution quickly rose above 42 °C within 3 min (Figure 2B), which is below the maximum permissible exposure (1 W cm^−2^) of 1064 nm laser [40,41]. We also investigated the concentration-dependent photothermal behavior of CM-LFPP. The temperature of the CM-LFPP solution rapidly increased under 1064 nm laser irradiation (0.6 W cm^−2^) at different concentrations (15.6, 31.3, 62.5, and 125 μg mL^−1^), with the highest temperature variation observed at a concentration of 125 μg mL^−1^ (Figure 2C). In contrast, the temperature of the PBS solution only increased by about 4 °C under the same 1064 nm laser irradiation, confirming the concentration-dependent photothermal effect of CM-LFPP (Figure 2D). To evaluate the photothermal conversion efficiency of CM-LFPP, we recorded the temperature variation curve using a 1064 nm laser with both on and off states (Figure 2E). The system time constant (*τ_s_*) of CM-LFPP was determined by fitting the cooling time against the negative natural logarithm of driving force temperature (−ln (θ)). The photothermal conversion efficiency is estimated to be approximately 78.7%, which exceeds that of most reported photothermal agents (Table 1). These results indicate that CM-LFPP is an excellent agent for converting light into heat for cancer therapy.

### 3.3. In Vitro PA/MRI Dual-Modal Imaging Ability of CM-LFPP

To assess the dual-modal imaging capabilities of CM-LFPP, we conducted measurements of its magnetic resonance and PA contrast abilities. The PA signal is intrinsically linked to the photothermal effect; therefore, we evaluated the PA imaging ability of CM-LFPP due to its exceptional photothermal conversion efficiency. We measured the PA signals of CM-LFPP at various concentrations ranging from 0 mg mL^−1^ to 1 mg mL^−1^ under 1064 nm laser irradiation. As depicted in Figure 3A, PA signals were detected even at low concentrations (0.063 mg mL^−1^) and increased with increasing concentration, demonstrating the concentration-dependent PA property of CM-LFPP. Furthermore, a strong linear relationship was observed between the concentration of CM-LFPP and the PA signal intensity, confirming its efficacy as an excellent contrast agent for PA imaging.

Fe_2_O_3_ nanoparticles with a size larger than 5 nm are commonly used as T2 contrast agents. Therefore, we recorded T2-weighted images (T2WI) of the CM-LFPP and ferumoxytol particles (FP). As shown in Figure 3B, the T2 images of the CM-LFPP and FP solutions darkened as the concentrations increased, indicating that they are both good T2 contrast agents for MRI. Additionally, we determined the corresponding transverse relaxivity (r2) to be 53.1 mM^−1^ s^−1^ and 48.7 mM^−1^ s^−1^ for FP and CM-LFPP, respectively, which were higher than the previous report and the commercial Resovist medium [46]. The similar r2 value demonstrated that the biometric modification had nearly no effect on the T2 imaging ability of the FP. These results demonstrate that CM-LFPP can be utilized as a dual-modal contrast agent for PA imaging and MRI.

### 3.4. In Vitro Biocompatibility

Before potential biological applications, the cytotoxicity of CM-LFPP was investigated using an MTT assay to measure cell cytotoxicity. The bEnd.3 and PC-3 cell lines, which represent normal and cancer cells, respectively, were exposed to varying concentrations of CM-LFPP for 12 and 24 h. The results indicated that cell cytotoxicity remained nearly unchanged at the tested concentrations, demonstrating the excellent biocompatibility of CM-LFPP (Figure 4A,B). Additionally, even at a concentration high up to 2 mg mL^−1^ after 24 h of incubation, the hemolysis rate was less than 10%, further confirming its biosafety (Figure 4C). Importantly, cancer cells were effectively eliminated under 1064 nm laser irradiation in the presence of CM-LFPP, while cell cytotoxicity remained stable under 1064 nm laser irradiation without CM-LFPP (Figure 4D). These findings suggest that CM-LFPP, as a PA/MRI dual-modal contrast agent, maybe a promising tool for imaging-guided cancer PTT.

### 3.5. In Vivo NIR-II PA Imaging

Based on the promising in vitro performance, we conducted a further evaluation of the in vivo NIR-II PA imaging and tumor-targeting ability of the biomimetic CM-LFPP using PC-3 tumor-bearing mice. We utilized a commercial 3D PA imaging system (Tomowave LOIS-3D) to record the NIR-II PA signals of the tumor-bearing mice after intratumor injection of LFPP and CM-LFPP at different time intervals. Before probe injection, low PA signals were observed under 1064 nm laser irradiation due to the weak absorption of endogenous contrasts in the NIR-II region (Figure 5A). However, the PA signals significantly increased in both the LFPP-treated group and the CM-LFPP-treated group at 2 h post-intratumor injection. Quantitative analysis demonstrated high signal-to-background ratios (SBR) of the LFPP-treated group and CM-LFPP-treated group, up to ~42.3 and ~50.2, respectively, confirming the excellent PA imaging capacity of the probes (Figure 5C). Additionally, the PA signals remained stable for more than 12 h post intratumor injection, indicating that the probes could remain in the tumor area for an extended period. Interestingly, the PA intensity in the CM-LFPP-treated group remained at ~70% at 48 h post-injection, while the PA signal in the LFPP-treated group only remained at ~40%, indicating that the biomimetic probes had a longer tumor retention time, possibly due to their active-targeting ability. To confirm this, we monitored the in vivo PA signals of the tumor after intravenous administration of LFPP and CM-LFPP at different time points. The PA signals increased gradually and reached a maximum at 24 h after the injection of CM-LFPP, while the PA signals reached a maximum at 6 h after the administration of LFPP. It further confirmed that CM-LFPP had a longer retention time compared to LFPP (Figure 5B). Moreover, the SBR of the tumor region reached ~30.2 (24 h post-injection) in CM-LFPP-treated mice, which was 5 times higher than that of LFPP-treated mice (6.1, 6 h post-injection) (Figure 5D). These results demonstrated that the biomimetic probes could effectively accumulate in the tumor site, demonstrating their excellent active tumor-targeting ability. These findings suggest that CM-LFPP could be a promising candidate for non-invasive and accurate diagnosis of PC tumors.

### 3.6. In Vivo MRI

To further evaluate the effectiveness of CM-LFPP as a contrast agent with high T2 relaxivity, in vivo MRI was performed on PC-3 tumor-bearing mice. T2-weight MRI imaging was conducted before and after the injection of the contrast agent. Our findings indicate that intratumoral administration of LFPP and CM-LFPP resulted in a significant decrease in T2 signal intensity in the tumor regions, as indicated by white-dotted lines in Figure 6A. This suggests that both contrast agents have good MRI capacity. Quantitative analysis further confirmed the decreased T2 signal intensity post-injection compared to pre-injection for both contrast agents, as shown in Figure 6B. Furthermore, we assessed the active targeting ability of the contrast agents by intravenous injection of LFPP and CM-LFPP, respectively. Before injection, the MRI signals of the tumor regions in both groups were similar. However, at 24 h post-injection, the tumor region of the CM-LFPP-treated mice exhibited significantly darker MRI signals compared to the LFPP-treated mice, as illustrated in Figure 6C,D. This indicates that CM-LFPP has superior MRI capacity and active tumor-targeting ability compared to LFPP.

### 3.7. In Vivo PTT

Ascribing to the high photothermal conversion efficiency of CM-LFPP, we investigated its in vivo photothermal antitumor capacity in PC-3 tumor-bearing mice. Infrared thermal images of the mice were captured after intratumor injection of PBS and CM-LFPP under 1064 nm laser irradiation. The temperature of the tumor region in the CM-LFPP-treated group increased rapidly from 30.2 to 50.6 °C after only 2 min of irradiation and continued to increase with irradiation time. After 6 min of 1064 nm laser irradiation, the temperature reached 57.2 °C, which was sufficient to destroy the tumor cells [47]. Furthermore, we conducted histological (H&E staining) and immunohistochemical (TUNEL) analyses to further verify the antitumor effect after PTT. No observable signs of tumor cell destruction were detected in the control groups, including PBS-treated, CM-LFPP-treated, and laser-treated mice. However, cell apoptosis was observed in the experiment group (laser + CM-LFPP) in the tumor tissue (Figure 7C), which was consistent with the TUNEL immunofluorescence results (Figure 7D), indicating that CM-LFPP is an excellent photothermal agent for tumor therapy.

### 3.8. In Vivo Biosafety Evaluation of the CM-LFPP

To investigate the in vivo biosafety of CM-LFPP, the theranostic agents were administered intravenously to healthy mice at a dose of 2 mg kg^−1^, which was four times higher than the imaging dose. Blood samples were collected before (0 days) and after (1 day, 4 days, and 7 days) injection of CM-LFPP, and various routine blood parameters such as white blood cell count (WBC^#^), lymphocyte count (Lym^#^), monocyte count (Mon^#^), etc. were measured. As shown in Table 2, all measured blood routine indices were within the normal range for healthy untreated mice at all time points tested, indicating that the injection of CM-LFPP did not induce an abnormal immune or inflammatory response in the mice. In addition, representative indices of liver function (ALT, AST, ALB, ALP, and TP) and kidney function (CREA, UA, and BUN) remained normal throughout the experimental period, suggesting that CM-LFPP did not cause liver or kidney injury (Figure 8A–H). To further investigate the histocompatibility of CM-LFPP, mice treated with PBS and CM-LFPP were randomly selected for haematoxylin and eosin (H&E) staining of major organs 7 days after injection. The results showed no detectable pathological abnormalities or inflammation in the major organs, indicating the good histocompatibility of CM-LFPP (Figure 8I). These results demonstrate that CM-LFPP has excellent biosafety, making it suitable for further biological applications.

## 4. Conclusions

In this study, we have developed a biomimetic cell membrane camouflaged Fe_2_O_3_ nanoclusters incorporated into polypyrrole (CM-LFPP) that exhibits strong NIR-II absorption and high T2 relaxivity. The CM-LFPP has active tumor-targeting capabilities and can be used as a theranostic agent for NIR-II PA/magnetic resonance dual-modal imaging-guided PTT of prostate cancer. The CM-LFPP has a high photothermal conversion efficiency of up to 78.7% at 1064 nm, making it a suitable candidate for PA imaging and MRI contrast agents due to the excellent photothermal properties and intrinsic properties of the encapsulated Fe_2_O_3_ nanoclusters. Additionally, the CM-LFPP demonstrates exceptional active tumor-targeting capabilities, resulting in high sensitivity in vivo NIR-II PA imaging with a high SBR ratio (~30.2) and long retention time (48 h) after intravenous injection. The tumor accumulation of the CM-LFPP was also confirmed by in vivo MRI. Furthermore, the biocompatible CM-LFPP demonstrated excellent photothermal antitumor efficacy under safe power density. This research provides a promising theranostic agent for imaging-guided tumor therapy, offering a design strategy for nanomaterials with satisfactory multimodal imaging capacity and good active tumor-targeting ability.

## Data Availability

The data that support the findings of this study are available from the corresponding author upon reasonable request.

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
