# Peer review of "Biomimetic Theranostic Agents with Superior NIR-II Photoacoustic and Magnetic Resonance Imaging Performance for Targeted Photothermal Therapy of Prostate Cancer"

_pharmaceutics, 2023, doi:10.3390/pharmaceutics15061617_

Round 1

Reviewer 1 Report

Liu et al. have presented the preparation and in vitro and in vivo performance of polypyrrole-coated Fe2O3 nanoclusters camouflaged with prostate cancer membrane fragments as theranostic agents in prostate cancer therapy. These nanoparticles feature remarkable MRI and photoacustic imaging performance and photothermal properties, combined with significant tumor targeting ability. The research work is surely interesting, and deserves publication, but a few aspects need to be improved.

1- The Introduction section should be improved by including some recent references to research work on cancer cell membrane-coated theranostics, specifically those developed for MRI and photoacustic imaging/photothermal therapy.

2- In the Materials and Methods section, the Materials paragraph is missing, as well as the description of the hemolytic test and the viability test performed under different treatment conditions (results reported in Fig. 4D).

3- In section 2.1 the amount of pyrrole used for the PPY coating and the pore size of the filter used for final CM-LFPP extrusion are not reported.

4- In section 2.1 the Authors mention "cell membrane proteins were extracted". Actually, according to mentioned ref. 33, the cells underwent some extrusion steps followed by centrifugation, obtaining, as again stated in ref. 33, "cell membrane fragments", which are likely made of proteins and phospholipids. Please clarify.

5- How did the Authors verify that nucleic acid material from the parent cells is not present in the cell membrane dispersion?

6- Did the Authors verify which proteins from the parent cell membranes are present in the cell membrane coating of CM-LFPP?

7- I would suggest to always express the concentration of nanoparticle dispersons in mcg/mL; also, the Authors should explain how they measured the weight of nanoparticles in the dispersions; finally, in line 151 they expressed the amount of nanoparticles as mmol Fe/L, but it should also be expressed as nanoparticle mass/vol, as throughout the rest of the paper.

8- In Figure 1B the dimension corresponding to the scale bar is not reported.

9- In Figure 2 no error bars are shown: were the measurements carried out only once?

Minor amendments: acronym PPY appears in line 179, but should be reported when "polypyrrole" is used for the first time, i.e. in line 69; in the caption of Fig. 4 "bEnd.3" is miswritten.

Author Response

Liu et al. have presented the preparation and in vitro and in vivo performance of polypyrrole-coated Fe2O3 nanoclusters camouflaged with prostate cancer membrane fragments as theranostic agents in prostate cancer therapy. These nanoparticles feature remarkable MRI and photoacustic imaging performance and photothermal properties, combined with significant tumor targeting ability. The research work is surely interesting, and deserves publication, but a few aspects need to be improved.

Response: Thanks for your valuable suggestions.

  1. The Introduction section should be improved by including some recent references to research work on cancer cell membrane-coated theranostics, specifically those developed for MRI and photoacustic imaging/photothermal therapy.

Response: Thanks for your suggestion. We have added some recent references in the revised manuscript (Ref.30-34).

  1. Li, S.; Jiang, W.; Yuan, Y.; Sui, M.; Yang, Y.; Huang, L.; Jiang, L.; Liu, M.; Chen, S.; Zhou, X. Delicately Designed Cancer Cell Membrane-Camouflaged Nanoparticles for Targeted 19F MR/PA/FL Imaging-Guided Photothermal Therapy. ACS Appl Mater Interfaces, 2020, 12, 51: 57290-57301.
  2. Pan, Y.; Zhu, Y.; Xu, C.; Pan, C.; Shi, Y.; Zou, J.; Li, Y.; Hu, X.; Zhou, B.; Zhao, C.; Gao, Q.; Zhang, J.; Wu, A.; Chen, X.; Li, J. Biomimetic Yolk-Shell Nanocatalysts for Activatable Dual-Modal-Image-Guided Triple-Augmented Chemodynamic Therapy of Cancer. ACS Nano, 2022, 16, 11: 19038-19052.
  3. Lu, J.; Mao, Y.; Feng, S.; Li, X.; Gao, Y.; Zhao, Q.; Wang, S. Biomimetic smart mesoporous carbon nanozyme as a dual-GSH depletion agent and O2 generator for enhanced photodynamic therapy. Acta Biomater, 2022, 148: 310-322.
  4. Fang, X.; Wu, X.; Li, Z.; Jiang, L.; Lo, W. S.; Chen, G.; Gu, Y.; Wong, W. T. Biomimetic Anti-PD-1 Peptide-Loaded 2D FePSe3 Nanosheets for Efficient Photothermal and Enhanced Immune Therapy with Multimodal MR/PA/Thermal Imaging. Adv Sci , 2020, 8, 2: 2003041.
  5. Li, H.; Liu, Y.; Huang, B.; Zhang, C.; Wang, Z.; She, W.; Liu, Y.; Jiang, P. Highly Efficient GSH-Responsive "Off-On" NIR-II Fluorescent Fenton Nanocatalyst for Multimodal Imaging-Guided Photothermal/Chemodynamic Synergistic Cancer Therapy. Anal Chem, 2022, 94, 29: 10470-10478.

  1. In the Materials and Methods section, the Materials paragraph is missing, as well as the description of the hemolytic test and the viability test performed under different treatment conditions (results reported in Fig. 4D).

Response: Thanks for pointing out this. The materials paragraph, as well as the description of the hemolytic test and the cell cytotoxicity test performed under different treatment conditions were added in the revised manuscript.

Materials: Iron chloride (98%), oleic acid (90%), 1-octadecene (90%), PVP40000, and pyrrole (99%) were obtained from Sigma-Aldrich. The 1,2-dipalmitoyl-sn-glycero-3-phosphocholine (DPPC), 1,2-dioleoyl-sn-glycero-3-phosphocholine (DOPC) and cholesterol were received from Avanti. All commercial reagents were used as received without additional purification. Ultrapure water was used for the experiments.

Cell culture and cell cytotoxicity: To evaluate the cytotoxicity of CM-LFPP, both bEnd.3 cell lines and prostate cancer PC-3 cell lines were used in an MTT assay. Normal and cancer cells were incubated at 37°C with 5% CO2 in DMEM and seeded onto 96-well plates at a density of 5 × 103 cells/well (100 µL). After removal of the original medium, DMEM containing various concentrations of CM-LFPP (0, 15.6, 31.3, 62.5, 125, 250, 500 µg mL-1) were incubated with the cells. The cytotoxicity of CM-LFPP was measured using the MTT assay after 12 and 24 hours of incubation. In addition, the PC-3 cell cytotoxicity of four groups including PBS-treated, CM-LFPP-treated, laser-treated, and CM-LFPP + laser- treated were also measured using the MTT assay. For CM-LFPP + laser- treated group, PC-3 cells incubated with CM-LFPP (250 µg mL-1) for 4 hours at 37℃, washed with PBS and irradiated under 1064 nm laser (0.6 W cm-2, 6 minutes).

Hemolysis test: Blood was collected from BALB/c mice weighing 20-22 g using anticoagulation tubes. The blood was washed with PBS (5 mL) and then centrifuged (2000 rpm, 3 minutes). Different concentrations of CM-LFPP were added to the samples, and a positive control with double-distilled water and a negative control with PBS were included. All samples were incubated at 37°C for 1 hour, and images were taken and the absorption at 540 nm was measured for each group. The hemolysis rate was calculated using the following formula. Hemolysis rate = (A540-experimental group - A540-sample - A540-negative control)/( A540-positive control - A540-negative control) × 100%

  1. In section 2.1 the amount of pyrrole used for the PPY coating and the pore size of the filter used for final CM-LFPP extrusion are not reported.

Response: Thanks for your reminding. We have added the amount of pyrrole (51.7 μL, 0.74 mmol) used for the PPY coating and the pore size (200 nm) of the filter used for final CM-LFPP extrusion. The information was also added in the revised manuscript.

  1. In section 2.1 the Authors mention "cell membrane proteins were extracted". Actually, according to mentioned ref. 33, the cells underwent some extrusion steps followed by centrifugation, obtaining, as again stated in ref. 33, "cell membrane fragments", which are likely made of proteins and phospholipids. Please clarify.

Response: We are sorry for the unclear description. The cell membrane fragments which including proteins and phospholipids were obtained according to the protocol. We have modified this description in the revised manuscript.

  1. How did the Authors verify that nucleic acid material from the parent cells is not present in the cell membrane dispersion?

Response: Thanks for your good question. The cell membrane fragments were obtained by collection the sediment after centrifugation, while the nucleic acid material were disperse well in the supernatant. By using this processing method, we could remove nucleic acid material from the parent cells.

  1. Did the Authors verify which proteins from the parent cell membranes are present in the cell membrane coating of CM-LFPP?

Response: Thanks for your good question. The cell membrane is rich in membrane proteins, making it difficult to determine which specific proteins are encapsulated on CM-LFPP. We appreciate the interesting suggestion provided by the reviewer, and we will conduct relevant research in the upcoming experiments.

  1. I would suggest to always express the concentration of nanoparticle dispersions in mcg/mL; also, the Authors should explain how they measured the weight of nanoparticles in the dispersions; finally, in line 151 they expressed the amount of nanoparticles as mmol Fe/L, but it should also be expressed as nanoparticle mass/vol, as throughout the rest of the paper.

Response: Thanks for your good suggestion. The weight of nanoparticles in the dispersions was measured according to this procedure. 1 mL of the CM-LFPP solution was lyophilized using vacuum freezing and drying technology, and the resulting powder was weighed to determine the concentration of the nanoparticles. To compare the T2 relaxivity of CM-LFPP with the previous reports, we measured the amount of iron (Fe) element using ICP-MS after digesting the CM-LFPP. Typically, the unit "mM-1s-1" is used for T2 relaxivity, so we did not use mass concentration.

  1. In Figure 1B the dimension corresponding to the scale bar is not reported.

Response: Thanks for pointing out this. The scale bars in Figure 1B indicate 50 nm. We have added it in the revised manuscript.

  1. In Figure 2 no error bars are shown: were the measurements carried out only once?

Response: Thanks for your question. The measurements were carried out three times. However, we only captured the representative thermal images once and their quantitative data were shown in the figure. Therefore, we didn’t give the error bars. 

  1. Minor amendments: acronym PPY appears in line 179, but should be reported when "polypyrrole" is used for the first time, i.e. in line 69; in the caption of Fig. 4 "bEnd.3" is miswritten.

Response: Thanks for pointing out this. We have modified these mistakes in the revised manuscript.

Reviewer 2 Report

The work shows high photothermal conversion efficiency of up to 78.7%, high signal-to-background ratio of ~30.2 for NIR-II photoacoustic imaging, good magnetic resonance imaging ability with a T2 relaxivity of up to 48.7 s-1mM-1, and remarkable actively target tumors effect of CM-LFPP NPs. I suggest its publication after the revisions below:

  1. How did the lipids encapsulation and biomimetic cell membrane modification enable CM-LFPP to actively target tumors separately? Is there any mechanism?
  2. The full name is lacking for the first appearance of some short names (CM-LFPP, PVP) in main text. In vivo/in vitro should be italic.
  3. Performance of photothermal conversion efficiency and T2 relaxivity should be compared with previous literature reports of other types of Fe2O3 or magnetic iron oxide NPs.
  4. Scale bars should be labeled with length in Fig 1B, H.
  5. What’s the PDI of NPs in Fig 1I?
  6. Please add the UV-Vis spectrum of single Pyrrole in Fig 1K.
  7. Why is the start temperature different between Fig 2E and Fig 2B/D?
  8. I suggest re-testing T2-weighted image of CM-LFPP one more time; maybe the signal at 0.5 mM Fe may increase, and the r2 will be slightly improved.
  9. In hemolytic test, why are there obvious hemolytic appearances in photo while the hemolytic rate didn’t change a lot?
  10. How to do quantitative analysis for PAI and MRI? Please write the details in “2. Materials and Methods” part."

Minor editing of English language required

Author Response

The work shows high photothermal conversion efficiency of up to 78.7%, high signal-to-background ratio of ~30.2 for NIR-II photoacoustic imaging, good magnetic resonance imaging ability with a T2 relaxivity of up to 48.7 s-1mM-1, and remarkable actively target tumors effect of CM-LFPP NPs. I suggest its publication after the revisions below:

Response: Thanks for your valuable suggestions.

  1. How did the lipids encapsulation and biomimetic cell membrane modification enable CM-LFPP to actively target tumors separately? Is there any mechanism?

Response: Thanks for your question. Biomimetic NPs utilize cell membrane fragments with specific proteins or ligands to enable self-recognition. The targeting mechanism relies on the interaction between natural membrane proteins and target cells, achieved by embedding cell membrane proteins into the NPs (1-3). In addition, encapsulation of lipids can enhance the fluidity of the biomimetic cell membrane, thereby improving the tumor-targeting capacities (4-5).

References

  1. Fang, R. H.; Kroll, A. V.; Gao, W.; Zhang, L. Cell Membrane Coating Nanotechnology. Adv. Mater. 2018, 30, 1706759,
  2. Chen, Z.; Zhao, P.; Luo, Z.; Zheng, M.; Tian, H.; Gong, P.; Gao, G.; Pan, H.; Liu, L.; Ma, A.; Cui, H.; Ma, Y.; Cai, L. Cancer Cell Membrane-Biomimetic Nanoparticles for Homologous-Targeting Dual-Modal Imaging and Photothermal Therapy. ACS Nano 2016, 10, 10049– 10057
  3. Sun, H.; Su, J.; Meng, Q.; Yin, Q.; Chen, L.; Gu, W.; Zhang, P.; Zhang, Z.; Yu, H.; Wang, S.; Li, Y. Cancer-Cell-Biomimetic Nanoparticles for Targeted Therapy of Homotypic Tumors. Adv. Mater. 2016, 28, 9581– 9588
  4. Liu, L.; Pan, D.; Chen, S.; Martikainen, M.-V.; Kårlund, A.; Ke, J.; Pulkkinen, H.; Ruhanen, H.; Roponen, M.; Käkelä, R., Xu, W.; Wang, J.; Lehto, V.-P. Systematic Design of Cell Membrane Coating to Improve Tumor Targeting of Nanoparticles. Nat Commun, 2022, 13, 1: 6181.
  5. Niu, Q.; Gao, J.; Zhao, K.; Chen, X.; Lin, X.; Huang, C.; An, Y.; Xiao, X.; Wu, Q.; Cui, L.; Zhang, P., Wu, L.; Yang, C. Fluid Nanoporous Microinterface Enables Multiscale-Enhanced Affinity Interaction for Tumor-Derived Extracellular Vesicle Detection. Proc Natl Acad Sci USA, 2022, 119, 44: e2213236119.

  1. The full name is lacking for the first appearance of some short names (CM-LFPP, PVP) in main text. In vivo/in vitro should be italic.

Response: Thanks for your question. The full name of the abbreviations were given in the revised manuscript. The phrase “In vivo/in vitro” was also changed to italic.

  1. Performance of photothermal conversion efficiency and T2 relaxivity should be compared with previous literature reports of other types of Fe2O3 or magnetic iron oxide NPs.

Response: Thanks for your good suggestion. We have compared the photothermal conversion efficiency and T2 relaxivity of Fe-based nanoparticles (Table R1). It can be observed that the photothermal conversion efficiency (PCE) of the obtained CM-LFPP was much higher than the previous literature reports, while the T2 relaxivity was lower than these magnetic iron oxide NPs.

Table R1. Comparison of the photothermal conversion efficiency (PCE) and T2 relaxivity of Fe-based nanoparticles

Fe-based agents

T2 relaxivity

PCE

Laser

References

Fe3O4@C

384 mM1·s1

21.7%

750 nm

1

MNC@ND‐DOX

82.1 mM1·s1

37.2%

808 nm

2

Fe@Fe3O4

156 mM1·s1

20%

808 nm

3

FPCH-DOX

171.8 mM1 s1

36.2%

808 nm

4

IONP@shell-in-shell

105 mM1 s 1

28.3%

1064 nm

5

CM-LFPP

48.7 mM-1 s-1

78.7%

1064 nm

This work

References

  1. Lu, A.H.; Zhang, X.Q.; Sun, Q; Zhang, Y; Song, Q.W.; Schuth, F.; Chen, C.Y.; Cheng, F.; Chen, C.Y. Precise synthesis of discrete and dispersible carbon-protected magnetic nanoparticles for efficient magnetic resonance imaging and photothermal therapy. Nano Res, 2016, 9, 5: 1460-1469.
  2. Li, Y.; Kong, J.; Zhao, H.; Liu, Y. Synthesis of Multi-Stimuli Responsive Fe3O4 Coated with Diamonds Nanocomposite for Magnetic Assisted Chemo-Photothermal Therapy. Molecules, 2023, 28, 4:1784.
  3. Zhou, Z.; Sun, Y.; Shen, J.; Wei, J.; Yu, C.; Kong, B.; Liu, W.; Yang, H.; Yang, S.; Wang, W. Iron/iron oxide core/shell nanoparticles for magnetic targeting MRI and near-infrared photothermal therapy. Biomaterials, 2014, 35, 26:7470-7478.
  4. Lin, X.; Song, X.; Zhang, Y.; Cao, Y.; Xue, Y.; Wu, F.; Yu, F.; Wu, M.; Zhu, X. Multifunctional theranostic nanosystems enabling photothermal-chemo combination therapy of triple-stimuli-responsive drug release with magnetic resonance imaging. Biomater Sci, 2020, 8, 7:1875-1884.
  5. Tsai, M. F.; Hsu, C.; Yeh, C. S.; Hsiao, Y. J.; Su, C. H.; Wang, L. F. Tuning the Distance of Rattle-Shaped IONP@Shell-in-Shell Nanoparticles for Magnetically-Targeted Photothermal Therapy in the Second Near-Infrared Window. ACS Appl Mater Interfaces, 2018, 10, 2:1508-1519.

  1. Scale bars should be labeled with length in Fig 1B, H.

Response: Thanks for your reminding. The scale bars in Figure 1B and H indicate 50 nm. We have added it in the figure captions of revised manuscript.

  1. What’s the PDI of NPs in Fig 1I?

Response: Thanks for your question. The PDI of the CM-LFPP was measured to be 0.241 ± 0.043. We have added it in the revised manuscript.

  1. Please add the UV-Vis spectrum of single Pyrrole in Fig 1K.

Response: Thanks for your suggestion. The UV-Vis spectrum of single Pyrrole was recorded and added in the revised Fig.1K. It can be observed that there was nearly no absorption of single Pyrrole (Figure R1a). In addition, we have also prepared polypyrrole in the absence of PVP40000, which couldn’t disperse in the water (Figure R1b).

Figure R1. a) The absorption of the FP, Pyrrole, FPP, and CM-LFPP; b) the photo of polypyrrole prepared in the absence of PVP40000.

  1. Why is the start temperature different between Fig 2E and Fig 2B/D?

Response: Thanks for your question. The starting temperature of Figure 2E was 27.5℃, whereas the starting temperature of Figures 2B/D was 23℃. The slight difference of 4.5℃ between the starting temperatures can be attributed to the varying environmental temperatures on the two days of the experiment.

  1. I suggest re-testing T2-weighted image of CM-LFPP one more time; maybe the signal at 0.5 mM Fe may increase, and the r2 will be slightly improved.

Response: Thanks for your suggestion. We have re-tested the T2-weighted image and T2 mapping of the FP and CM-LFPP. As shown in Figure R2, the r2 was slightly improved, which was still similar to that of FP. It showed that the biometric modification had nearly no effect on the T2 imaging ability of the FP.

Figure R2. a) T2-weighted images (T2WI) and T2 mapping of the FP and CM-LFPP; b) the linear fitting of inverse T2 vs the concentration of Fe in FP and CM-LFPP.

  1. In hemolytic test, why are there obvious hemolytic appearances in photo while the hemolytic rate didn’t change a lot?

Response: Thanks for your good question. The CM-LFPP solution appears black due to its wide absorption range of 530 nm to 1100 nm, which intensifies as the concentration of CM-LFPP increases. Consequently, it is difficult to identify hemolysis based solely on the solution's color. To minimize the impact of solution absorption, the hemolysis rate can be calculated using the provided formula, which explains the consistent hemolysis rate. It is inferred that the darkened colors in the photograph are a result of the presence of CM-LFPP. In addition, the details of hemolysis experiment were added in the revised manuscript.

Hemolysis rate = (A540-experimental group - A540-sample - A540-negative control)/( A540-positive control - A540-negative control) × 100%

Hemolysis experiment: Blood was collected from BALB/c mice weighing 20-22 g using anticoagulation tubes. The blood was washed with PBS (5 mL) and then centrifuged (2000 rpm, 3 minutes). Different concentrations of CM-LFPP were added to the samples, and a positive control with double-distilled water and a negative control with PBS were included. All samples were incubated at 37°C for 1 hour, and images were taken and the absorption at 540 nm was measured for each group. The hemolysis rate was calculated using the following formula. Hemolysis rate = (A540-experimental group - A540-sample - A540-negative control)/( A540-positive control - A540-negative control) × 100%

  1. How to do quantitative analysis for PAI and MRI? Please write the details in “2. Materials and Methods” part."

Response: Thanks for your suggestion. We evaluated the photoacoustic imaging capability of CM-LFPP using a commercial 3D photoacoustic imaging system (Tomowave LOIS-3D). The quantitative photoacoustic imaging data was obtained through a plugin installed in the 3D slice, developed by the company utilizing ImageJ. The MRI was quantitatively analyzed by measuring the tumor signals of various slices using the MRI system's software. We have included these details in the revised manuscript.

Comments on the Quality of English Language

Minor editing of English language required

Response: Thanks for your suggestion. The language is modified in the revised manuscript.

Reviewer 3 Report

There are a few issues with this report on the ability of a preparation of Fe2O3 to create local heating effects upon irradiation. It is pointed out that the PSA test identifies prostate cancer but does not ‘localize’ the tumor in the prostate. Since the usual therapy involves either surgical removal or ionizing radiation to destroy the tumor site, is there any need to identify the region of the lesion? There may be a case for a more specific targeting approach but this is never revealed.

If prostate cancer can be dealt with by ionizing radiation nor surgery, what is the need for another approach? There may such a need, but if so what is it?

The introduction mentions that late-stage prostate cancer has a lower survival rate. This is because of the tendency of this disease to metastasize, e.g., to bones.  PTT will be of little use once this has occurred.  The Introduction appears to be proposing an approach to detection of early-stage prostate cancer. There is no comparison with PSA results, so it is difficult to see what advantages might be obtained with CM-LFPP as a diagnostic procedure. The protocol calls for ‘intratumoral’ injection, but this assumes the presence of a tumor. How would this procedure be used in a typical patient where the presence of a tumor is unknown?

The absorbance profile of the agent (Fig. 1 panel K) shows a broad peak that includes the region produced by a 1064 nm laser. Use of the MTT assay is initially indicated to reflect cytotoxicity, but the authors later begin to use the term ‘viability’. Oleinick has shown that there is a poor correlation between MTT data and clonogenic determinations of viability. Use of the term ‘viability’ is to be avoided unless tests for cell proliferation are employed. The MTT assay detects the activity of some mitochondrial dehydrogenases at a single time-point. Nothing else.

It might be argued that once diagnosed, early prostate cancer that has not spread beyond the prostate could be treated with a photothermal process. This could be more selective than  surgical or ionizing radiation procedures, both of which are accompanied by significant adverse effects. Use of MRI could be useful in tumor localization. It seems unlikely that the CM-LFPP approach is going to replace the PSA test for large-scale preliminary diagnostic procedures. Only clinical trials will reveal whether this approach will be more successful and associated with fewer adverse effects than current therapy for prostate tumor that is confined to the prostate.    

Author Response

  1. There are a few issues with this report on the ability of a preparation of Fe2O3 to create local heating effects upon irradiation. It is pointed out that the PSA test identifies prostate cancer but does not ‘localize’ the tumor in the prostate. Since the usual therapy involves either surgical removal or ionizing radiation to destroy the tumor site, is there any need to identify the region of the lesion? There may be a case for a more specific targeting approach but this is never revealed.

Response: Thanks for your question. Conventional blood tests for prostate-specific antigen (PSA) has played important roles in qualitative screening. However, it usually needs to conduct MRI and biopsy for patients with abnormal PSA index to decrease the misdiagnosis rate before surgical removal or ionizing radiation. As mentioned by the reviewer, surgical removal and ionizing radiation are important methods to treat prostate cancer. Accurate surgery and targeted radiation therapy can effectively minimize potential side effects. Thus, the precise localization of the cancerous area through imaging techniques holds significance. In this study, we developed biomimetic cell membrane-modified theranostic agents and verified their excellent targeting capabilities in a mouse subcutaneous tumor model using both photoacoustic imaging and magnetic resonance imaging.

  1. If prostate cancer can be dealt with by ionizing radiation nor surgery, what is the need for another approach? There may such a need, but if so what is it?

Response: Thanks for your question. Although ionizing radiation and surgery have proven to be effective treatments for prostate cancer, there is a need to explore alternative approaches to further enhance treatment efficacy. One promising avenue is the utilization of imaging-guided photothermal therapy as a complementary treatment modality. In a clinical trial conducted by Naomi J. Halas's group, magnetic resonance-ultrasound fusion imaging-guided photothermal therapy mediated by gold-silica nanoshells was successfully employed. Focal laser ablation was achieved in 94% (15/16) of patients, with no significant differences observed in the International Prostate Symptom Score or Sexual Health Inventory for Men after treatment (PNAS, 2019, 116(37):18590-18596). Continual research and development of new treatment methods are necessary to improve treatment outcomes and enhance the well-being of prostate cancer patients.

  1. The introduction mentions that late-stage prostate cancer has a lower survival rate. This is because of the tendency of this disease to metastasize, e.g., to bones. PTT will be of little use once this has occurred. The Introduction appears to be proposing an approach to detection of early-stage prostate cancer. There is no comparison with PSA results, so it is difficult to see what advantages might be obtained with CM-LFPP as a diagnostic procedure. The protocol calls for ‘intratumoral’ injection, but this assumes the presence of a tumor. How would this procedure be used in a typical patient where the presence of a tumor is unknown?

Response: Thanks for your valuable question. We agree with the reviewer's assessment that focal photothermal therapy is not a suitable treatment for metastatic prostate cancer tumors. Imaging techniques are used as complementary methods to PSA testing to confirm the diagnosis and locate the tumor. We also concur that intratumoral injection is not practical for patients with unknown tumors. However, the focus of this manuscript is to provide a strategy for the development of theranostic agents and to validate their performance in imaging-guided tumor therapy. The targeting and therapeutic efficacy of the obtained CM-LFPP were validated through photoacoustic and magnetic resonance imaging, as well as laser irradiation in a subcutaneous tumor mouse model. The experimental results demonstrate the feasibility of this design strategy.

  1. The absorbance profile of the agent (Fig. 1 panel K) shows a broad peak that includes the region produced by a 1064 nm laser. Use of the MTT assay is initially indicated to reflect cytotoxicity, but the authors later begin to use the term ‘viability’. Oleinick has shown that there is a poor correlation between MTT data and clonogenic determinations of viability. Use of the term ‘viability’ is to be avoided unless tests for cell proliferation are employed. The MTT assay detects the activity of some mitochondrial dehydrogenases at a single time-point. Nothing else.

Response: We appreciate the reviewer's comment regarding the use of the term "viability" in relation to the MTT assay. We agree that the MTT assay is primarily used to assess cytotoxicity rather than overall cell viability, especially considering the poor correlation between MTT data and clonogenic determinations of viability as demonstrated by Oleinick. We acknowledge the importance of using precise terminology in scientific research. To avoid any confusion, we have changed the “viability” to “cytotoxicity” in the revised manuscript.

  1. It might be argued that once diagnosed, early prostate cancer that has not spread beyond the prostate could be treated with a photothermal process. This could be more selective than surgical or ionizing radiation procedures, both of which are accompanied by significant adverse effects. Use of MRI could be useful in tumor localization. It seems unlikely that the CM-LFPP approach is going to replace the PSA test for large-scale preliminary diagnostic procedures. Only clinical trials will reveal whether this approach will be more successful and associated with fewer adverse effects than current therapy for prostate tumor that is confined to the prostate.

Response: Thanks for your valuable comments. Compared with other tumor treatments, photothermal therapy (PTT) exhibits several advantages, including reduced side effects, minimal invasiveness, and precise control. Naomi J Halas’s group has conducted magnetic resonance-ultrasound fusion imaging-guided photothermal therapy in a clinical trial, which has verified the concept. Additionally, we agree with the reviewer's comments that the CM-LFPP approach is not able to replace the PSA test for large-scale preliminary diagnostic procedures. PSA, as a blood-based screening method, plays a crucial role in the screening of prostate cancer. Imaging methods, on the other hand, serve primarily as supplementary tools for further screening of individuals with abnormal PSA levels.

Round 2

Reviewer 1 Report

I would suggest the Authors to include all the answers given to the Reviewers in the manuscript. And strongly advise to conduct a verification of the absence of nucleic acids in the biologic material obtained from cancer cells.